# Tobacco Smoking Is a Strong Predictor of Failure of Conservative Treatment in Hinchey IIa and IIb Acute Diverticulitis—A Retrospective Single-Center Cohort Study

**DOI:** 10.3390/medicina59071236

**Published:** 2023-07-02

**Authors:** Valentina Murzi, Eleonora Locci, Alessandro Carta, Tiziana Pilia, Federica Frongia, Emanuela Gessa, Mauro Podda, Adolfo Pisanu

**Affiliations:** 1Department of Emergency and Acute Care, Emergency Surgery Unit, Cagliari University Hospital, Cagliari-Monserrato, 09042 Cagliari, Italy; 2Department of Surgical Science, University of Cagliari, 09124 Cagliari, Italy

**Keywords:** acute diverticulitis, diverticular abscess, tobacco smoking, risk factors, conservative treatment

## Abstract

*Background and Objectives:* Therapeutic management of patients with complicated acute diverticulitis remains debatable. The primary objective of this study is to identify predictive factors for the failure of conservative treatment of Hinchey IIa and IIb diverticular abscesses. *Materials and Methods:* This is a retrospective cohort study that included patients hospitalized from 1 January 2014 to 31 December 2022 at the Emergency Surgery Department of the Cagliari University Hospital (Italy), diagnosed with acute diverticulitis complicated by Hinchey grade IIa and IIb abscesses. The collected variables included the patient’s baseline characteristics, clinical variables on hospital admission, abscess characteristics at the contrast-enhanced CT scan, clinical outcomes of the conservative therapy, and follow-up results. Univariable and multivariable logistic regression models were used to identify prognostic factors of conservative treatment failure and success. *Results:* Two hundred and fifty-two patients diagnosed with acute diverticulitis were identified from the database search, and once the selection criteria were applied, 71 patients were considered eligible. Conservative treatment failed in 25 cases (35.2%). Univariable analysis showed that tobacco smoking was the most significant predictor of failure of conservative treatment (*p* = 0.007, OR 7.33, 95%CI 1.55; 34.70). Age (*p* = 0.056, MD 6.96, 95%CI −0.18; 0.99), alcohol drinking (*p* = 0.071, OR 4.770, 95%CI 0.79; 28.70), platelets level (*p =* 0.087, MD −32.11, 95%CI −0.93; 0.06), Hinchey stage IIa/IIb (*p =* 0.081, OR 0.376, 95%CI 0.12; 1.11), the presence of retroperitoneal air bubbles (*p =* 0.025, OR 13.300, 95%CI 1.61; 291.0), and the presence of extraluminal free air at a distance (*p =* 0.043, OR 4.480, 95%CI 1.96; 20.91) were the other variables possibly associated with the risk of failure. In the multivariable logistic regression analysis, only tobacco smoking was confirmed to be an independent predictor of conservative treatment failure (*p =* 0.006; adjusted OR 32.693; 95%CI 2.69; 397.27). *Conclusion:* The role of tobacco smoking as a predictor of failure of conservative therapy of diverticular abscess scenarios highlights the importance of prevention and the necessity to reduce exposure to modifiable risk factors.

## 1. Introduction

Conservative treatment of complicated diverticular disease with abscess formation represents one of the most insidious challenges for emergency abdominal surgeons and healthcare systems worldwide [1,2]. The incidence of diverticular disease has undergone an exponential increase in recent years, both in Western countries (maximum prevalence in the United States, Europe, and Australia) [3] and in developed Eastern countries, such as Japan [3,4].

Different classification systems for complicated acute diverticulitis exist. The Hinchey classification, formulated in 1978, has been the most used in the international literature over the last three decades and divides acute diverticulitis into four stages based on the severity of the disease [5]. In 1997, Sher et al. modified the classification to distinguish between pericolic abscesses (I), distant abscesses amendable for percutaneous drainage (IIA), and complex abscesses associated with a possible fistula (IIb). This classification can be used to counsel the implementation of CT-guided percutaneous drainage of abscesses as an effective strategy for treatment [6]. In 2005, Kaiser et al. formulated another modified Hinchey classification based on specific CT scan findings [7,8].

Therapeutic strategy choices for patients with complicated acute diverticulitis remains open for debate. The choice of treatment differs according to the size of the abscess itself. An initial non-operative approach is suggested for small abscesses (≤4–5 cm). Intravenous antibiotic therapy, with in-hospital monitoring to detect a possible progression to free perforation and septic complications, is considered a safe and effective strategy in most cases, with a 20% failure rate and a fatality rate of 0.6% [1]. When the abscess is >5 cm, antibiotics may fail to achieve adequate concentration within the abscess, resulting in increased failure rates. A diameter greater than 4–5 cm represents the reasonable cut-off between antibiotic treatment alone and its association with ultrasound (US)- or CT-guided percutaneous drainage. If percutaneous drainage is not feasible or not available, patients with large abscesses can undergo an antibiotic challenge with close in-hospital monitoring. Alternatively, a colonic resection with a Hartmann procedure or primary anastomosis is required [1,9].

### Aim of the Study

There has been a wealth of research in observational and randomized controlled studies focusing on treating Hinchey I, III, or IV disease [10,11,12,13,14]. Conversely, little focus has been placed on the cohort of patients presenting with acute diverticulitis complicated by pericolic or pelvic Hinchey IIa-IIb abscesses. Despite the well-established role of abscess size as a predictive factor for failure, an open question remains regarding recognizing the cut-off of 4–5 cm in diameter as a threshold value for deciding the therapeutic approach. Even small abscesses may progress to complications with a failure of conservative therapy, and even when resolved, they may be associated with a high recurrence rate [15]. Some studies conducted in recent years have found possible correlations between some parameters and the failure of conservative treatment. In the study by Ahmadi et al., the C-reactive protein (CRP) level and its trajectory during the initial 48 h of admission could predict the need for intervention in acute diverticulitis patients being managed conservatively [16]. Al-Masrouri et al. found that the readmission rate for treatment failure after an episode of acute diverticulitis managed nonoperatively was 6.6%, and having had a previous episode of complicated diverticulitis was the strongest risk factor for treatment failure [17]. In a systematic review, Lee et al. found that non-operative management’s failure rate was 44.4%. The rate of relapse at 30 days following non-operative management was 18.9%. Distant abscesses were associated with significantly increased recurrence rates compared with pericolic abscesses. The recurrence rate following non-operative management was 25.5% at the follow-up of 38 months [18]. In 2021, a meta-analysis by Fowler et al. found that failure rates following non-operative management of acute diverticulitis complicated by abscess did not significantly decrease over the past three decades [19]. Given the increasing global incidence of this condition, it is crucial to understand other factors that may contribute to the failure of antibiotic therapy and/or percutaneous drainage to help establish better treatment strategies. This would help resolve the acute symptomatic event, reduce discomfort, prevent relapses, and improve quality of life, while avoiding unnecessary procedures, reducing hospitalization length, and optimizing healthcare resources.

## 2. Materials and Methods

### 2.1. Study Design

This was a retrospective cohort study that included patients admitted from 1 January 2014 to 31 December 2022 at the Emergency Surgery Unit of the Cagliari University Hospital (Italy) with a diagnosis of acute diverticulitis complicated with grade IIa and IIb Hinchey diverticular abscess. Inclusion criteria were as follows: patients ≥18 years old with a Hinchey IIa or IIb diverticular disease documented with contrast-enhanced CT, and patients who underwent conservative treatment with antibiotic therapy and possible US or CT-guided percutaneous drainage of the abscess. In addition, patients diagnosed with uncomplicated acute diverticulitis or complicated acute diverticulitis other than IIa and IIb grades and patients undergoing emergency surgical treatment on admission were excluded. The patients were identified by searching the electronic register of the Surgical Emergency Department admissions. This study was conducted under the principles of the Declaration of Helsinki and was developed and presented according to Strengthening the Reporting of Observational Studies in Epidemiology (STROBE) [20]. Study approval was granted by the Institutional Review Board of the Cagliari State University Hospital (Italy), whereas ethical review was waived for this study due to its retrospective, non-interventional design.

### 2.2. Study Variables

Patient’s medical records were examined, and the data were extrapolated and collected in a database created with Microsoft Excel^©^ version 16.74 (Microsoft Corporation, Redmond, WA, USA, 2021). The collected variables concerned the patient’s baseline characteristics and medical history [age, sex, body mass index (BMI), Charlson’s Comorbidity Index, comorbidities, smoking and alcohol consumption], clinical variables on hospital admission [vital parameters, blood tests, length of stay in the Emergency Department, timing of hospitalization, number of previous episodes of acute diverticulitis, *Clostridium difficile* infection, duration of symptoms before admission], abscess characteristics at the contrast-enhanced CT scan [number of abscesses, maximum diameter in mm, presence of air bubbles inside the abscess and/or in the retroperitoneum and extraluminal free air, presence of pelvic free fluid, Hinchey staging grade IIa/IIb], variables regarding conservative therapy [type of antibiotic, use of US or CT-guided percutaneous drainage], clinical outcomes [failure of conservative treatment, reason for failure, management strategies in case of failure of conservative treatment, days passed from the start of conservative therapy and failure], and post-hospitalization outcomes [recurrence within and/or after 30 days, recurrence characteristics].

### 2.3. Study Objectives

The primary objective of this study was to identify predictive factors for the failure of conservative treatment of Hinchey IIa and IIb diverticular abscesses, considering the baseline and clinical characteristics of the patients, the characteristics of the abscesses on CT imaging, the management strategy, and the resulting clinical outcomes.

### 2.4. Statistical Analysis

Baseline characteristics of the study population were expressed as absolute numbers and relative frequency measures for qualitative variables. Mean and standard deviation (SD) or median and standard error (SE)/interquartile range (IQR) were used for quantitative variables. Differences between groups for qualitative variables were determined using the *X^2^* or Fisher’s exact test. In contrast, comparisons of quantitative variables were performed using Student’s *t*-test (variables with parametric distribution) and Mann’s U test—Whitney (nonparametric distribution). Univariable and multivariable logistic regression models were used to identify prognostic factors of conservative treatment failure and success. Variables yielding *p* values < 0.1 from the single-variable analysis of associations were added to a stepwise prediction model based on their predictive value, and the goodness of fit of the binary logistic regression model was assessed by determining the pseudo-*R*^2^ (*R*^2^ by Negelkerke and *R*^2^ by Cox and Snell). The strength of the association between an identified risk factor from univariable and multivariable analyses for treatment failure and success was determined by calculating odds ratios (OR) and adjusted odds ratios (aOR) with 95% confidence intervals (95% CI). A *p* value < 0.05 (two-tailed) was considered statistically significant. All statistical analyses were performed using Jamovi computer software (the Jamovi project 2022; Jamovi Version 2.3).

## 3. Results

Two hundred and fifty-two patients with acute diverticulitis were identified from the database search, and once the selection criteria were applied, 71 patients were considered eligible (Figure 1). The baseline characteristics of the study cohort at the time of admission are shown in Table 1.

The active principle of the antibiotics administered for conservative treatment of diverticular abscesses in the study cohort.

### 3.1. Hospitalization Data

A median time between the onset of symptoms and hospitalization of three days (IQR 7.0) was reported in the study cohort, while the median time spent in the Emergency Department was 285 min (IQR 262). Hospitalization occurred in 83.1% (n. 59) on a weekday and 16.9% (n. 12) on weekends. The hospitalization time was included in 35.7% (n. 25) of the cases in the time slot between 12:01 and 18:00, in 34.3% (n. 24) between 18:01 and 23:59, in 22.7% (n. 16) between 0:00 and 06:00, and 7.1% (n. 5) between 06:01 and 12:00 (data missing for one patient); 81.7% (n. 58) of patients had no history of acute diverticulitis, whereas 8.4% (n. 6) reported more than one episode. Vital signs recorded at admission showed fever in 21.1% of patients (n. 15), tachycardia in 31% of patients (n. 22), and a mean systolic blood pressure of 132 mmHg (SD ± 17.4). The blood tests at admission showed a mean leukocyte count of 14.0 × 10^3^/µL (SD ± 4.5); median platelet count was 260 × 10^3^/µL (IQR 130); 19.7% of patients (n. 14) had anemia (Hb < 12 g/dL), whereas 23.9% (n. 17) had renal failure (creatinine >1.2 mg/dL). The median CRP value was 107.0 mg/L (IQR 130.0), and procalcitonin on admission was 0.17 ng/mL (IQR 0.38) (Table 1). Duration of symptoms before hospital admission > three days was reported in 56.6% (n. 26) and 72.0% (n. 18) patients in the conservative treatment success and failure groups, respectively (Table 2).

### 3.2. Data Relating to the Abscess Characteristics on CT Scan

On abdominal CT scans, 85.9% (n. 61) of patients had only one abscess, 9.8% (n. 7) had two abscesses, and 4.2% (n. 3) had more than two abscesses. The median maximum diameter of the abscess was 45.0 mm (IQR 39.0); 47.9% (n. 34) of the abscesses had air bubbles in their context whereby in 44.1% (n. 15) a single air bubble was found, and in 55.9% (n. 19) more than one air bubble was reported at the CT scan. In 12.7% (n. 9) of the cases, the presence of free pelvic fluid was reported; in 11.3% (n. 8) of cases, the CT scan showed distant extraluminal air, and in 2.8% (n. 2) of cases, retroperitoneal air bubbles were reported. According to the Hinchey classification, 38.1% (n. 27) patients had stage ΙΙa, and 61.9% (n. 44) had ΙΙb grade (Table 1).

### 3.3. Management Data

In eight cases (11.3%), CT-guided percutaneous drainage was associated with antibiotic therapy. The culture test report was available for all eight cases. The most detected bacteria were *Escherichia coli* (50% of the available sample, n. 4) and *Klebsiella pneumoniae* (25%, n. 2). *Staphylococcus aureus*, *Staphylococcus epidermidis*, *Streptococcus anginosus*, *Morganella morganii*, *Enterobacter cloacae*, *Proteus mirabilis*, *Candida albicans* were also found only once each.

### 3.4. Clinical Outcomes

Conservative treatment failed in 25 cases (35.2%). Failure of the conservative treatment when the antibiotic therapy was associated with the use of percutaneous drainage was reported in two cases (Figure 2). The main reason for the failure of conservative management was the lack of size reduction of the abscess on the follow-up CT scan (68%; n. 17). The second reason was the occurrence of diffuse peritonitis at the physical examination (20%; n. 5). Additional reasons are shown in Figure 3. The median time between the initiation of conservative treatment and its failure was 12 days (IQR 8.0). For patients who underwent surgery after failure of conservative treatment, the primary approach was left hemicolectomy with primary anastomosis in 56% (n. 14) of patients, followed by left hemicolectomy with open abdomen (28%, n. 7), and colonic resection with Hartmann’s procedure (16%, n. 4). Complications occurred in seven (9.8%) patients. The grade of surgical complications, evaluated according to the Clavien-Dindo classification, was ΙΙ in 42.7% (n. 3) of cases. In contrast, grade I and a ΙΙΙb complications were found in 28.6% (n. 2) of the cases. The median length of hospital stay was 14 days (IQR 13.0). The length of hospital stay for cases with failure and success of non-operative management was 25.0 ± 8.2 days for patients with failure and 10.9 ± 5.5 days for patients with the success of conservative treatment (*p* < 0.001, 95%CI −17.4; −10.8), with a mean difference of −14.1. No abscess recurrence occurred after conservative treatment at 30-day follow-up after discharge. Conversely, in two cases, a recurrence was reported after 30 days, presenting as an abscess. As shown in Figure 4, starting in 2017, the rate of successful conservative treatment started increasing in conjunction with the implementation of CT-guided percutaneous drainage of diverticular abscesses larger than five centimeters.

### 3.5. Analysis of the Predictive Factors of Conservative Treatment Failure

Univariable analysis showed that tobacco smoking was the most significant predictor of failure of conservative treatment (*p =* 0.007, OR 7.33, 95%CI 1.55; 34.70). In addition, age (*p =* 0.056, MD 6.96, 95%CI −0.18; 0.99), alcohol drinking (*p =* 0.071, OR 4.770, 95%CI 0.79; 28.70), platelets level (*p =* 0.087, MD −32.11, 95%CI −0.93; 0.06), Hinchey stage IIa/IIb (*p =* 0.081, OR 0.376, 95%CI 0.12; 1.11), the presence of retroperitoneal air bubbles (*p =* 0.025, OR 13.300, 95%CI 1.61; 291.0), and the presence of extraluminal free air at distance (*p =* 0.043, OR 4.480, 95%CI 1.96; 20.91) were the other variables possibly associated with the risk of failure (*p* < 0.01) (Table 2). However, in the multivariable logistic regression analysis, only tobacco smoking was confirmed to be an independent predictor of conservative treatment failure (*p =* 0.006; adjusted OR 32.693; 95%CI 2.69; 397.27) (Table 3).

## 4. Discussion

With approximately one billion smokers, according to the World Health Organization (WHO), tobacco smoke is the greatest threat to health and the first risk factor for chronic diseases worldwide. Given the role of tobacco smoking and other environmental factors, such as diet, physical inactivity, and alcohol, on the pathogenesis, progression, complications, and risk of recurrence of diverticular disease, the reduction of exposure to tobacco smoking is necessary to decrease the impact of modifiable risk factors on the natural history of acute diverticulitis. It is also necessary to identify additional factors that can predict the failure of the conservative management of diverticular abscesses in order to release clear and shared guidelines on preventive strategies, therapeutic management (especially concerning the role of antibiotic therapy and percutaneous drainage for reducing the need for emergency surgery) of patients with complicated acute diverticulitis. This approach would save patients from unnecessary treatments, improve their quality of life, and decrease the use of healthcare resources.

The results of this study strengthen the evidence that tobacco smoking is associated with a higher incidence of complications in patients with acute diverticulitis [21] and a higher risk of conservative treatment failure in patients with diverticular abscesses [22]. Tobacco smoking can be a pathogenetic and prognostic factor through several mechanisms. Firstly, nicotine acts by inhibiting cytokines (IL-1 and TNF) and leukocytes, reduces collagen formation and causes an increase in intraluminal pressure with impaired blood flow to the colonic mucosa and increased tissue permeability. Such mechanisms can favor the evolution towards perforation and increase inflammation, favoring endothelial dysfunction and reducing the blood oxygen supply [23,24]. This correlates with the current hypothesis of the pathogenesis of diverticulitis on an ischemic basis, reinforcing its validity [25].

Additionally, tobacco smoking can alter the composition and diversity of the intestinal microbiome [26]. Finally, tobacco smoking also acts through its effect on the cardiocirculatory system, being a risk factor for hypertension, atherosclerosis, chronic renal failure, diabetes, and cerebrovascular disease. All of these are, as demonstrated by the studies by Niikura et al. [27] and Okamoto et al. [28], independent predictors for the onset of diverticular complications. Furthermore, not to be underestimated is that tobacco smoking has been associated with an increased rate of perforations in acute appendicitis and colorectal anastomotic leak in patients who underwent resection for colon cancer [29,30].

The results of this study were in line with the study published by Turunen et al., which demonstrated that tobacco smoking favors the development of complications of diverticulitis and is a significant determinant in the failure of conservative therapy. In this retrospective series of 261 patients operated on for diverticular disease at Helsinki University Central Hospital over five years, smokers underwent sigmoidectomy earlier in their life than non-smokers and had an increased rate of perforations and episodes of recurrent postoperative diverticulitis [31].

The role of age should be investigated. In our study, the mean age difference between patients for whom conservative treatment failed and those in which it was successful was 6.96 years, suggesting the hypothesis that failure correlates with age in the univariable analysis. However, based on the data available and this study’s design, it is impossible to definitively establish the assumption that failure correlates with age, rather than to a more aggressive clinical presentation in the acute episode [2], but a higher recurrence rate is valid. Our study is not powered by an adequate statistical analysis for the outcome recurrence. Within this context, the natural history of acute diverticulitis, the presentation of recurrent episodes following successful conservative therapy, as well as the role of the time factor between the acute presentation and hospitalization, mandates a reflection on the role of patient follow-up over time to prevent the onset of complications and to plan the most appropriate therapeutic intervention. In synthesis, a tailored, personalized approach based on a comprehensive and detailed assessment of the patient and his pathology is the key to increasing the success of treatments.

Significant results that emerged from the univariable analysis of the time elapsed between the presentation of symptoms and hospitalization led us to hypothesize the role of the precocity of conservative treatment. Data analysis showed that cases that met with success of conservative therapy presented at the emergency department after 3 ± 0.79 days from the onset of symptoms, while those that met with failure presented 5 ± 1.67 days later, with a mean difference of 3.54 days. This could be due to inflammation and virulent bacterial action in the progression of the disease, as well as, among smokers, a possible role of non-abstinence from smoking during the acute period. However, such hypotheses require evidence from further studies.

Staging according to the Hinchey classifications was found to be significant in the univariable analysis. This is reinforced by the analysis of the main diameter of the abscess on CT, highlighting how the conservative treatments that met with success were performed on smaller abscesses. These results confirm the current guidelines by the World Society of Emergency Surgery (WSES) [1], the European Association of Endoscopic Surgery (EAES), and the Society of American Gastrointestinal and Endoscopic Surgery (SAGES) [32], but also suggest the ability of a dimensional range as the main decision-making tool. A further observation that emerged is the variable trend of the success curve during the years we have examined. Of patients who experienced success with the conservative treatment, 9.86% underwent percutaneous drainage versus 1.41% in the failure group. Although this difference is not statistically significant, it is likely due to a small number of cases and imprecision errors; this is relevant and should be further investigated with more significant numbers at our center. The 2018 EAES and SAGES panel of experts recommended that percutaneous drainage be considered for abscesses > 4 cm, those that do not resolve on antibiotics, and/or in the presence of patient deterioration [32]. The first percutaneous drainage in our cohort was recorded in 2019, and 88% of drainages were placed in the last two years (data not shown). These data confirmed the increasingly recognized efficacy of percutaneous drainage as a resolutive treatment and bridging to help reduce inflammation and make the patient eligible for colonic resection with primary anastomosis. However, the role of percutaneous drainage in this context needs to be clarified. In the study by Elagili et al. [33], treatment outcomes in patients with diverticular abscesses larger than 3 cm in diameter were analyzed. In particular, 32 patients were initially treated with antibiotics alone and 114 with percutaneous drainage. The study found no significant differences between the drainage and antibiotic groups regarding the need for emergency surgery (18% vs. 25%). The authors suggested that, in selected patients, antibiotics alone without drainage can be implemented as an initial treatment, even in the case of large diverticular abscesses. 

Garfinkle et al. [34] evaluated the long-term safety of nonsurgical treatment of diverticular abscesses. Of the 73 patients in their retrospective study, 33 underwent percutaneous drainage and had a low incidence of subsequent emergency procedures (2.7% during the 62-month follow-up). Conversely, another retrospective study of 185 conservatively managed patients, of whom 31% were treated with percutaneous drainage, found that 28% required emergency surgery [35].

Based on the mentioned studies, the abscess size can be indicative, but it should not be the main element in choosing the treatment, as it does not correlate in an absolute way with a successful outcome. Percutaneous drainage is an option to consider in possible association with antibiotic therapy, although its success does not even eliminate the risk of future complications and recurrences. The therapeutic choice must be personalized according to the anamnestic and clinical characteristics of the patient. In this context, further studies are needed to understand all the parameters and risk factors that can predict the failure of conservative therapy in patients with diverticular abscess scenarios, leading to a better understanding of the disease and patient outcomes.

### Limitations

The results of this study must be interpreted with caution, given the small patient sample from a single center. Overall, 36 variables were tested, making the risk of multiple testing errors high. Moreover, the Hinchey classification was missing for nineteen patients, although abscess diameter was reported for every participant. An imprecision bias and a selection bias inherent in the observational and retrospective nature of the study itself can burden our findings. Future projects should be designed to verify our preliminary data in the context of a multicenter collaborative experience.

## 5. Conclusions

The role of tobacco smoking as an independent predictor of failure of conservative therapy of acute diverticulitis complicated by Hinchey ΙΙa and ΙΙb abscesses reported in this study highlighted the importance of prevention.

## Figures and Tables

**Figure 1 medicina-59-01236-f001:**
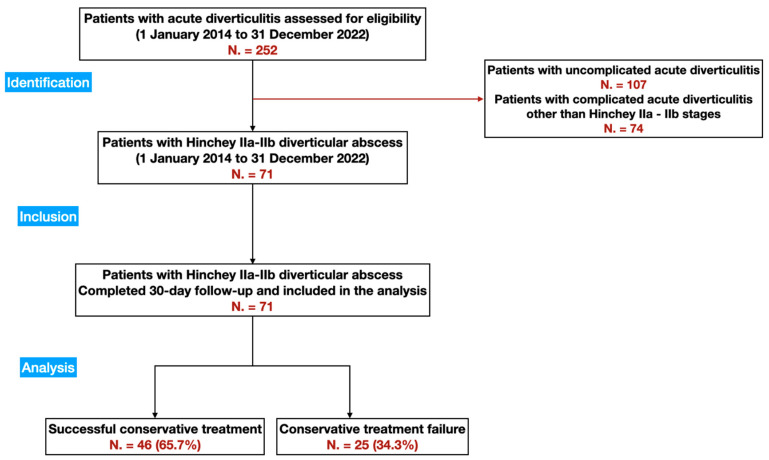
Study STROBE Flow Diagram.

**Figure 2 medicina-59-01236-f002:**
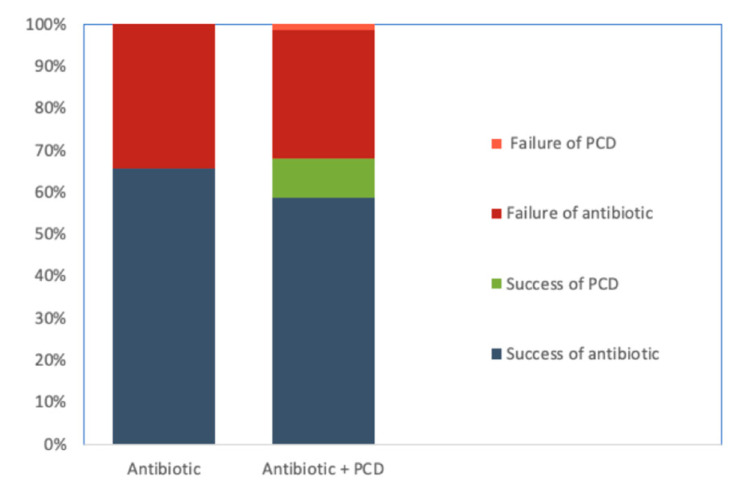
Outcomes of the conservative treatment.

**Figure 3 medicina-59-01236-f003:**
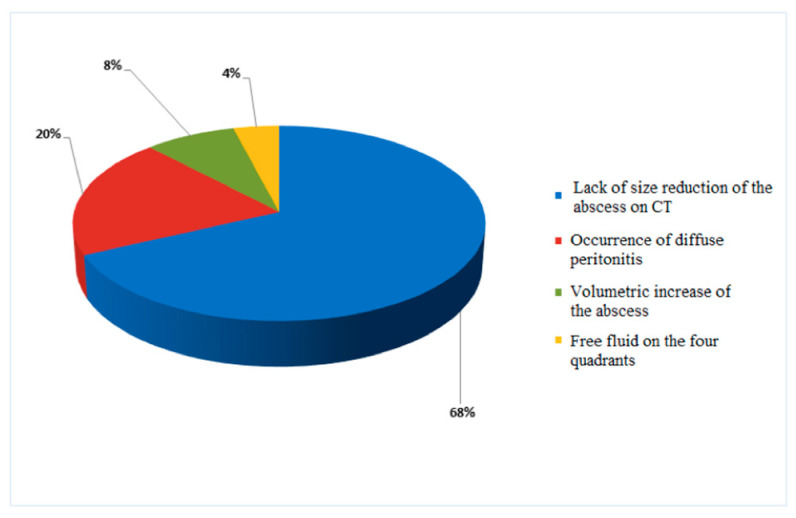
Reasons for conservative treatment failure.

**Figure 4 medicina-59-01236-f004:**
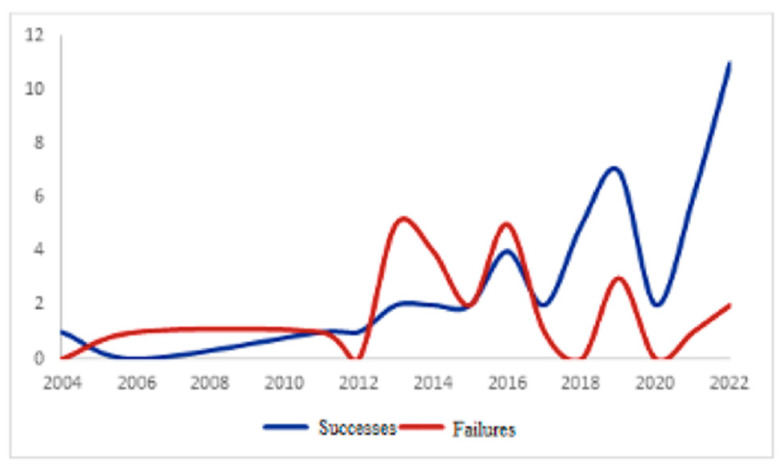
Trends in conservative treatment success and failure.

**Table 1 medicina-59-01236-t001:** Baseline characteristics of the study cohort.

Variable	Sample Size (Total Examined)	N.	%	Mean	Standard Deviation	Median	Interquartile Range
**Age (year)**	71	71		61.6	14.4	63.0	19.0
**Sex (Male)**	71	35	49.3				
**BMI (Kg/m^2^)**	35	35		26.5	4.4	26.2	5.3
**Charlson’s Comorbidity Index**	71	71		2.1	1.8	2.0	2.0
**Diabetes (N. %)**	71	5	7.0				
**Chronic renal failure (N. %)**	71	4	5.6				
**Therapy with corticosteroids (N. %)**	71	3	4.2				
**Heart failure (N. %)**	71	1	1.4				
**Obesity (N. %)**	35	7	9.7				
**Coagulopathy (N. %)**	71	1	1.4				
**Arterial hypertension (N. %)**	71	30	42.2				
**Chronic Obstructive Pulmonary Disease (COPD) (N. %)**	71	3	4.2				
**Coronaropathy (N. %)**	71	8	11.2				
**Renal failure (Creatinine > 1.2 mg/dL) (N. %)**	71	17	23.9				
**Smoking (N. %)**	71	11	15.4				
**Alcohol abuse (N. %)**	71	7	9.8				
**Leukocytes ×10^3^/μL**	71	71		14.0	4.5	13.6	5.8
**CRP mg/L**	41	41		130.0	92.3	107.0	130.0
**Anemia (Hb < 12 g/dL) (N. %)**	71	14	19.7%				
**Platelets × 10^3^/μL**	71	71		287	94.1	260	130
**Procalcitonin ng/mL**	43	17		6.7	24.1	0.2	0.4
**Fever (>38 °C) (N. %)**	71	15	21.1				
**Systolic blood pressure (mmHg)**	71	71		132	17.4	130	20.0
**Tachycardia (>100 bpm) (N. %)**	71	22	31.0				
**Patients with previous episodes of acute diverticulitis**	71	13	18.31				
**Diameter of the abscess on CT (mm)**	71	71		51.8	33.2	45.0	39.0
**Hinchey grade IIa (N. %)**	71	33	46.5				
**Hinchey grade IIb (N. %)**	71	38	53.5				
**Presence of air bubbles in the abscess (N. %)**	71	38	53.5				
**Presence of retroperitoneal air bubbles (N. %)**	71	2	3.1				
**Presence of extraluminal air (N. %)**	71	8	11.9				
**Presence of free pelvic fluid (N. %)**	71	9	13.4				
**CT-guided percutaneous drainage of the abscess (N. %)**	71	8	11.3				
**Time between the start of conservative therapy and failure (Days)**	71	25		13.6	8.4	12	8.0
**Duration of symptoms before admission (Days)**	71	71		5.8	6.6	3	7.0
**Time spent in the emergency department (Minutes)**	71	65		447	1054	285	262
**Time of admission (Hours)**	71						
** *06.01–12.00* **		5	7.1				
** *12.01–18.00* **		25	35.7				
** *18.01–23.59* **		24	34.3				
** *0.00–06.00* **		16	22.8				
**Day of the admission**							
** *Weekday* **		59	83.1				
** *Weekend* **		12	16.9				

**Table 2 medicina-59-01236-t002:** Results of the univariable analysis. Outcome: Failure of conservative antibiotic therapy +/− percutaneous drainage.

Variable	Success Group n = 46 (65.7%)	Failure Group n = 25 (34.3%)	*p* Value	Odds Ratio	Mean Difference	95% CI
**Age (years) ± mean difference (MD)**	63.9 ± 14.7	57.0 ± 13.1	0.056		6.96	−0.18; 0.99
**Sex (Male) n. (%)**	21 (45.6%)	14 (56.0%)	0.405	0.660		0.25; 1.76
**BMI (Kg/m^2^) ± mean difference (MD)**	26.5 ± 4.6	25.8 ± 3.5	0.615		0.74	−0.50; 0.85
**Charlson’s Comorbidity Index ± mean difference (MD)**	2.0 ± 0.2	2.0 ± 0.3	0.524		0.28	−0.34; 0.65
**Diabetes**	4 (8.7%)	1 (4.0%)	0.460	0.438		0.05; 4.14
**Chronic renal failure**	3 (6.5%)	1 (4.0%)	0.660	0.597		0.06; 6.06
**Corticosteroid therapy**	2 (4.3%)	1 (4.0%)	0.945	0.917		0.08; 10.60
**Heart failure**	-	1 (4.0%)	0.172	5.690		0.22; 142
**Obesity**	4 (8.7%)	3 (12.0%)	0.797	0.804		0.15; 4.25
**Coagulopathy**	-	1 (4.0%)	0.172	5.690		0.22; 145
**Hypertension**	22 (47.8%)	8 (32.0%)	0.197	0.513		0.19; 1.42
**Chronic Obstructive Pulmonary Disease (COPD)**	1 (2.2)	2 (8.0%)	0.244	3.910		0.34; 45.50
**Renal failure (N. %)**	12 (26.1%)	5 (20.0%)	0.566	0.708		0.21; 2.31
**Coronaropathy**	6 (13.0%)	2 (8.0%)	0.521	0.580		0.11; 3.11
**Tobacco Smoking**	3 (6.5%)	8 (32.0%)	0.007	7.330		1.55; 34.70
**Alcohol drinking**	2 (4.3%)	5 (20.0%)	0.071	4.770		0.79; 28.70
**Leukocytes × 10^3^/μL ± mean difference (MD)**	14.1 ± 4.9	13.7 ± 3.6	0.716		0.42	−0.40; 0.59
**CRP mg/L ± mean difference (MD)**	101.4 ± 17.8	144.1 ± 21.7	0.345		−15.30	−0.91; 0.52
**Anemia (Hb < 12 g/dL)**	13.0 ± 1.4	13.2 ± 1.7	0.665		−0.18	−0.60; 0.38
**Platelets × 10^3^/μL ± mean difference (MD)**	258 ± 13.8	299 ± 18.2	0.087		−32.11	−0.93; 0.06
**Procalcitonin ng/mL ± mean difference (MD)**	0.1 ± 6.6	0.3 ± 0.1	0.601		7.14	−1.20; 1.76
**Fever (>38 °C)**	19 (41.3%)	6 (24.0%)	0.662	1.301		0.40; 4.19
**Systolic blood pressure (mmHg) ± mean difference (MD)**	130 ± 16.5	135 ± 19.1	0.254		−1.52	−0.81; 0.22
**Tachycardia (>100 bpm)**	12 (26.1%)	10 (40.0%)	0.226	1.892		0.67; 5.32
**Number of previous episodes of acute diverticulitis (previous episodes)**	36 (78.2%) (0 episodes) 4 (8.7%) (1 episode) 6 (13.0%) (>1 episodes)	22 (88.0%) (0 episodes) 3 (12.0%) (1 episode) - (>1 episodes)	0.163			0.14; 2.41
**Diameter of the abscess on CT (mm) ± mean difference (MD)**	40.0 ± 4.3	50.0 ± 10.1	0.195		−14.43	−1.01; 0.13
**Number of abscesses on CT**	38 (82.6%) (1 abscess) 5 (10.9%) (2 abscesses) 3 (6.5%) (>2 abscesses)	23 (92.0%) (1 abscess) 2 (8.0%) (2 abscesses) - (>2 abscesses)	0.521	0.600		0.11; 3.16
**Hinchey stage IIa**	21 (45.6%)	6 (24.0%)	0.081	0.376		0.12; 1.11
**Hinchey stage IIb**	25 (54.4%)	19 (76.0%)				
**Presence of air bubbles in the abscess**	22 (47.8%)	12 (48.0%)	0.848	0.900		0.31; 2.64
**Number of air bubbles in the abscess**	24 (52.2%) (0 bubbles) 11 (23.9%) (1 bubble) 11 (23.9%) (>1 bubble)	8 (32.0%) (0 bubbles) 4 (16.0%) (1 bubble) 8 (32.0%) (>1 bubble)	0.839	1.320		0.44; 3.96
**Presence of retroperitoneal air bubbles**	-	2 (8.0%)	0.025	13.300		1.61; 291.0
**Presence of extraluminal air at distance**	3 (6.5%)	5 (20.0%)	0.043	4.480		1.96; 20.91
**Presence of free pelvic fluid**	6 (13.0%)	3 (12.0%)	0.890	1.110		0.25; 4.95
**CT-guided percutaneous drainage of the abscess**	7 (15.2%)	1 (4.0%)	0.153	0.232		0.03;2.01
**Duration of symptoms before hospital admission (>3 Days) (N. %)**	26 (56.5%)	18 (72.0%)	0.199	1.980		0.69; 5.65
**Time spent in the emergency department (minutes) ± mean difference (MD)**	303.0 ± 187.2	203.0 ± 40.8	0.111		276.68	−0.27; 0.79
**Day of the admission**				0.608	1.39	0.39; 4.95
** *Weekdays* **	39 (84.8%)	20 (80.0%)				
** *Weekends* **	7 (15.2%)	5 (20.0%)				

**Table 3 medicina-59-01236-t003:** Results of the multivariable logistic regression analysis.

Variables	*p*-Value	Adjusted Odds Ratio	95% CI
Age (years)	0.461	0.973	0.90; 1.05
**Tobacco smoking**	**0.006**	**32.693**	**2.69; 397.27**
Alcohol drinking	0.140	5.466	0.57; 52.29
Hinchey stage (2a/2b)	0.595	2.190	0.12; 39.53
Presence of retroperitoneal air bubbles	0.995	1.078	0.18; 12.37
Presence of extraluminal air at distance	0.428	2.919	0.20; 41.21

## Data Availability

Data supporting this study’s findings will be available upon request from the principal investigator (M.P.).

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
