# Peer review of "Tobacco Smoking Is a Strong Predictor of Failure of Conservative Treatment in Hinchey IIa and IIb Acute Diverticulitis—A Retrospective Single-Center Cohort Study"

_medicina, 2023, doi:10.3390/medicina59071236_

Round 1

Reviewer 1 Report

A study well conducted and written, although English needs some minor editing and polish. I have some concerns to address, which in my opinion, could improve the present work:

-       As more than 2/3 of patients from the initial search were excluded, a flowchart of patient selection could be useful.

-       In “results” section, many data are included in text and in table 1. This is redundant and unnecessary. Please, do not duplicate data.

-       Are data from blood test showed in table 1 from admission? Please, clarify this in M&M.

-       In table 1, please state unit of measurement for “Time between the start of conservative therapy and failure”.

-       Mean or median body temperature is useless. It could be better state number (%) of patients presenting fever/feverish. The same for heart rate, it could be more useful refer to tachycardia. The same for creatinine/renal failure, and even hemoglobin/anemia. Uni and multivariate analysis using these new variables should be carried out, and compared with current. 

-       In line 192, “For 18 patients, it was not possible to trace the 192 classification, according to Hinchey.” As multivariate analysis needs no missing data, How were classified these patients?

-       Figure 4 may be unnecessary as data are easily understandable.

-       Please, state clearly what was the length of hospital stays for cases with failure and success of non-operative management.

-       Regarding “the other variables associated with the risk of failure” is awkward to find a p > 0.05 and a 95% CI not including 1. Please check data, explain statistical findings or state only that those variables COULD be also associated.

-       In table 2, in qualitative variables, percentage must refer to the whole cases of the column (n. 46 for success group and n. 25 for failure group), not to the whole cases of the study cohort.

-        In table 2, in success group, 9.86% underwent percutaneous drainage vs. 1.41% in failure group. Although this difference is not statistically significant (probably due a small number of cases), this is relevant and should be explained and discussed, as a potential bias in patients undergoing drainage may be present.

-       In line 272, please clarify the following sentence “These results lead us to consider the possible importance of 271 the other data emerging from our study”.

-        In line 300, “Until 2017, the failure and success rates 300 were equivalent, whereas afterward, the trend was reversed”. Please, include complete data for this in the results section.

-       The conclusions section is too long. Please, summarize it.

A study well conducted and written, although English needs some minor editing and polish.

Author Response

REVISED MANUSCRIPT: RESPONSE TO REVIEWERS DOCUMENT.

Medicina (MDPI)

To the Editor-in-Chief of Medicina:

To the Reviewers of the Manuscript ID: medicina-2428354

"Tobacco smoking is a strong predictor of failure of conservative treatment in Hinchey IIa and IIb acute diverticulitis—a retrospective single-center cohort study."

Running Head: Predictors of failure of conservative treatment for diverticular abscesses.

Valentina Murzi MD, Eleonora Locci MD, Alessandro Carta MD, Tiziana Pilia MD, Federica Frongia MD, Emanuela Gessa MD, Mauro Podda MD FACS, and Adolfo Pisanu MD PhD.

Department of Emergency and Acute Care, Emergency Surgery Unit, Cagliari University Hospital, Cagliari, Italy & Department of Surgical Science, University of Cagliari, Cagliari, Italy.

Mauro Podda MD FACS

Department of Surgical Science, University of Cagliari

Emergency Surgery Unit, Policlinico Universitario “D. Casula”, Azienda Ospedaliero-Universitaria di Cagliari

SS 554, Km 4,500

09042, Monserrato (Italy)

Email: [email protected]; [email protected]

We thank the Editor-in-Chief, the Associate Editor, and the Reviewers for the opportunity to revise our manuscript # medicina-2428354, entitled "Tobacco smoking is a strong predictor of failure of conservative treatment in Hinchey IIa and IIb acute diverticulitis—a retrospective single-center cohort study." We are grateful for constructive comments and suggestions provided by the Reviewers, and we believe that their input has added to our manuscript. We have considered their comments and suggestions carefully and have tried to do our best to respond to the points raised from their review. Moreover, this will also allow us to clarify some important aspects of our study.

We have addressed all of the specific comments provided by the Reviewers and have made the requested changes or clarifications. As specified in the instructions provided by the Editor, we have highlighted the changes to our manuscript within the document by using red text.

Furthermore, we have created this document in order to respond "point by point" to the comments made by the Reviewers. Please refer to the text of the revised manuscript. Although this makes a rather lengthy response letter, we believe that it will provide the Editors and the Reviewers with the best explanation of the changes that we have made in response to the comments received.

We hope that the manuscript revisions and our accompanying responses will be sufficient to make our manuscript suitable for publication in Medicina (MDPI).

We shall look forward to hearing from you at your earliest convenience.

Yours, Sincerely

Mauro Podda MD FACS

Department of Surgical Science, University of Cagliari

Emergency Surgery Unit, Policlinico Universitario “D. Casula”, Azienda Ospedaliero-Universitaria di Cagliari

SS 554, Km 4,500

09042, Monserrato (Italy)

Email: [email protected]; [email protected]

Comments and Suggestions for Authors

Reviewer #1 A study well conducted and written, although English needs some minor editing and polish. I have some concerns to address, which in my opinion, could improve the present work:

- As more than 2/3 of patients from the initial search were excluded, a flowchart of patient selection could be useful. Authors: Thanks for your valuable suggestion on this point. A flowchart figure (Figure 1) has now been added to the manuscript. Figures have been renumbered accordingly.

"Two hundred and fifty-two patients with acute diverticulitis were identified from the database search, and once the selection criteria were applied, 71 patients were considered eligible [Figure 1]. Baseline characteristics of the study cohort at the time of admission are shown in Table 1." [Results].

- In “results” section, many data are included in text and in Table 1. This is redundant and unnecessary. Please, do not duplicate data. Authors: Data shown in Table 1 have now been deleted from the main text in order to avoid redundant and unnecessary reports. Thank you.

- Are data from blood test showed in Table 1 from admission? Please, clarify this in M&M. Authors: Thanks for your remark on this point. Baseline characteristics, including blood test showed in Table 1 were taken from the admission notes.

- In Table 1, please state unit of measurement for “Time between the start of conservative therapy and failure”. Authors: the unit of measurement for this specific variable was "days". The tables have now been double-checked for every unit of measurement. Thank you.

- Mean or median body temperature is useless. It could be better state number (%) of patients presenting fever/feverish. The same for heart rate, it could be more useful refer to tachycardia. The same for creatinine/renal failure, and even hemoglobin/anemia. Uni and multivariate analysis using these new variables should be carried out, and compared with current. Authors: Thanks for your useful suggestions on this point. Median body temperature has now been dichotomized in fever (yes/no), whereas hearth rate was categorized in tachycardia (yes/no), creatinine levels in renal failure (yes/no), and hemoglobin levels in anemia (yes/no). Similarly, the duration of symptoms before hospitalization was dichotomized in >3 or ≤ 3 days (Table 2). Univariable and multivariable analyses have been performed again accordingly. Although minor changes in the effects estimates were noted, the multivariable analysis confirmed that tobacco smoking was the only predictive factor of conservative treatment failure in patients with diverticular abscess.

- In line 192, “For 18 patients, it was not possible to trace the classification, according to Hinchey.” As multivariate analysis needs no missing data, How were classified these patients? Authors: Thanks for your insightful comment on this point. We performed a new database search on hospital notes and found the missing data. No data is now missing concerning the Hinchey classification (Table 1).

"On abdominal CT scan, 85.9% (n. 61) of patients had only one abscess, 9.8% (n. 7) had two abscesses, and 4.2% (n. 3) had more than two abscesses. The median maximum diameter of the abscess was 45.0 mm (IQR 39.0); 47.9% (n. 34) of the abscesses had air bubbles in their context: in 44.1% (n.15) a single air bubble was found and in 55.9% (n. 19) more than one air bubble was reported at the CT scan. In 12.7% (n. 9) of the cases, the presence of free pelvic fluid was reported; in 11.3% (n. 8), the CT scan showed distant extraluminal air, and in 2.8% (n. 2) patients, retroperitoneal air bubbles were reported. According to the Hinchey classification, 38.1% (n. 27) patients had stage ΙΙa, and 61.9% (n. 44) had ΙΙb grade [Table 1]."

- Figure 4 may be unnecessary as data are easily understandable. Authors: Figure 4 has now been withdrawn. Thank you.

- Please, state clearly what was the length of hospital stays for cases with failure and success of non-operative management. Authors: Thanks for your comment on this point. We performed a new data analysis and found that the length of hospital stay for cases with failure and success of non-operative management was 25.0 ± 8.2 days for patients with failure and 10.9 ± 5.5 days for patients with success of conservative treatment (p< 0.001, 95%CI -17.4; -10.8), with a mean difference of -14.1.

- Regarding “the other variables associated with the risk of failure” is awkward to find a p > 0.05 and a 95% CI not including 1. Please check data, explain statistical findings or state only that those variables COULD be also associated. Authors: Thanks for your comment on this point. We have double-checked our study results and found that variables for which a p value > 0.05 and 95%CI not including 1 were reported, were all quantitative variables. For mean differences the null value in statistics is 0, whereas for Odds ratio and Risk ratio it is 1. Reason for which, no amendments should be made to our results.

- In Table 2, in qualitative variables, percentage must refer to the whole cases of the column (n. 46 for success group and n. 25 for failure group), not to the whole cases of the study cohort. Authors: Data in the Table 2 have now been amended accordingly. Thank you.

- In Table 2, in success group, 9.86% underwent percutaneous drainage vs. 1.41% in failure group. Although this difference is not statistically significant (probably due a small number of cases), this is relevant and should be explained and discussed, as a potential bias in patients undergoing drainage may be present. Authors: thanks for your insightful comment on this point. We have now discussed this limitations, as follows:

"A further observation that emerged is the variable trend of the success curve during the years we have examined. Of patients who experienced success of the conservative treatment, 9.86% underwent percutaneous drainage versus 1.41% in the failure group. Although this difference is not statistically significant, probably due a small number of cases and imprecision error, this is relevant and should be further investigated with larger numbers at our center. The 2018 EAES and SAGES panel of experts recommended that percutaneous drainage should be considered for abscesses > 4 cm, those that do not resolve on antibiotics, and/or in the presence of patient deterioration [32]. The first percutaneous drainage in our cohort was recorded in 2019, and 88% of drainages were placed in the last two years (data not shown). These data confirmed the increasingly recognized efficacy of percutaneous drainage as a resolutive treatment and bridging to help reduce inflammation and make the patient eligible for colonic resection with primary anastomosis."

- In line 272, please clarify the following sentence “These results lead us to consider the possible importance of the other data emerging from our study”. Authors: the mentioned period has been withdrawn. It was used as transition period, but we recognize it was unnecessary and somewhat unclear. Thank you.

- In line 300, “Until 2017, the failure and success rates were equivalent, whereas afterward, the trend was reversed”. Please, include complete data for this in the results section. Authors: we double-checked our results following your suggestion. The amended the results section as follows:

"As shown in Figure 4, starting from 2017 the rate of successful conservative treatment started increasing in conjunction with the implementation of CT-guided percutaneous drainage of diverticular abscesses larger than five centimeters."

- The conclusions section is too long. Please, summarize it. Authors: Thanks for your advice. We have now summarized the conclusions as follows:

"The role of tobacco smoking as an independent predictor of failure of conservative therapy of acute diverticulitis complicated by Hinchey ΙΙa and ΙΙb abscesses reported in this study highlighted the importance of prevention."

Reviewer 2 Report

Dear Editor,

Thank you for the opportunity to review this manuscript.

This is a retrospective explorative study on the role of different predictive factors for failure of conservative treatment of Hinchey IIa and IIb diverticular abscesses on patients from a single institution. It is over all a well performed study and well written manuscript but in my opinion some information is lacking and I therefore have some questions and comments.

This is an exploratory study with 36 different variables making the risk for multiple testing errors high. This should be discussed under limitations.

If I understand correctly three different methods of selecting variables for the multivariable analysis was used. One was selection based on previous knowledge and two were mathematical (based on p-values and based on pseudo-R2 values). It would help the reader to know which variables were chosen based on previous knowledge and how this was motivated. It would also be interesting to know the motivation for using methods for selecting variables based on both p-values and pseudo-R2 values. Both these methods have similar drawbacks (e.g. adjusting results on the specific data sample, multiple hypothesis testing and so on) and while pseudo-R2 p is a better at estimating each variable’s impact on the model as a whole I don’t see any advantages of adding p-values as a basis for variable selection.

The layout of the table is difficult to read.

Was there missing data for any other variable than Hinchey classification? How was missing data treated in the analyses?

No significant difference was found in age between those with failed treatment and those with successful treatment. This might be a power issue since it was close to being significant.  The risk for recurrence was not examined. How can these results confirm the hypothesis that failure correlates with age due not to a more aggressive clinical presentation in the acute episode but a higher recurrence rate?

Author Response

Reviewer #2 Comments and Suggestions for Authors

This is a retrospective explorative study on the role of different predictive factors for failure of conservative treatment of Hinchey IIa and IIb diverticular abscesses on patients from a single institution. It is over all a well performed study and well written manuscript but in my opinion some information is lacking and I therefore have some questions and comments.

This is an exploratory study with 36 different variables making the risk for multiple testing errors high. This should be discussed under limitations. Authors: Thank you for your insightful comment on this point. We have modified the limitations section as follows:

"The results of this study must be interpreted with caution, given the small patient sample from a single center. Overall, 36 different variables were tested, making the risk for multiple testing errors high. Moreover, the Hinchey classification was missing for nineteen patients, although abscess diameter was reported for every participant. An imprecision bias and a selection bias inherent in the observational and retrospective nature of the study itself can burden our findings. Future projects should be designed to verify our preliminary data in view of a multicenter collaborative experience."

If I understand correctly three different methods of selecting variables for the multivariable analysis was used. One was selection based on previous knowledge and two were mathematical (based on p-values and based on pseudo-R2 values). It would help the reader to know which variables were chosen based on previous knowledge and how this was motivated. Authors: we have now amended the methods section, and the results of the multivariable analysis accordingly, stating that "Univariable and multivariable logistic regression models were used to identify prognostic factors of conservative treatment failure and success. Variables yielding P values <0.1 from the single-variable analysis of associations were added to a stepwise prediction model based on their predictive value, denoted by the pseudo-R2 (R2 by Negelkerke and R2 by Cox & Snell). The strength of the association between an identified risk factor from univariable and multivariable analyses for treatment failure and success was determined by calculating odds ratios (OR) and adjusted odds ratios (aOR) with 95% confidence intervals (95% CI). A P value <0.05 (two-tailed) was considered statistically significant. All statistical analyses were performed using Jamovi computer software (the Jamovi project 2022; Jamovi Version 2.3).

In summary, only variables that reached a P value < 0.1 at the univariable analysis entered the final multivariable model.

It would also be interesting to know the motivation for using methods for selecting variables based on both p-values and pseudo-R2 values. Both these methods have similar drawbacks (e.g. adjusting results on the specific data sample, multiple hypothesis testing and so on) and while pseudo-R2 p is a better at estimating each variable’s impact on the model as a whole I don’t see any advantages of adding p-values as a basis for variable selection. Authors: Thanks for your comment on this point. We selected the variables for entering the multivariable model based on P values <0.1 and then checked the pseudo-R2 to assess the goodness of fit.

We have now amend the results section as follows:

"Univariable and multivariable logistic regression models were used to identify prognostic factors of conservative treatment failure and success. Variables yielding P values <0.1 from the single-variable analysis of associations were added to a stepwise prediction model based on their predictive value, and the goodness of fit of the binary logistic regression model was assessed by determining the pseudo-R2 (R2 by Negelkerke and R2 by Cox & Snell). The strength of the association between an identified risk factor from univariable and multivariable analyses for treatment failure and success was determined by calculating odds ratios (OR) and adjusted odds ratios (aOR) with 95% confidence intervals (95% CI). A P value <0.05 (two-tailed) was considered statistically significant. All statistical analyses were performed using Jamovi computer software (the Jamovi project 2022; Jamovi Version 2.3)."

The layout of the table is difficult to read. Authors: The layout of the Tables has now been modified. Thank you.

Was there missing data for any other variable than Hinchey classification? How was missing data treated in the analyses? Authors: Thanks for your insightful comment on this point. We have now double-checked our hospital database and the radiology system. No missing data for the Hinchey classification are now reported. Missing data are still present for variables such as CRP, procalcitonin, and BMI. Columns having null values were deleted.

No significant difference was found in age between those with failed treatment and those with successful treatment. This might be a power issue since it was close to being significant. Authors: Thank you for your comment on this point, that allow us to provide the reader with some further clarifications. Age had a P value < 0.1 in the univariable analysis, and was then considered in the multivariable model where, however, was found not statistically significant.

The risk for recurrence was not examined. How can these results confirm the hypothesis that failure correlates with age due not to a more aggressive clinical presentation in the acute episode but a higher recurrence rate? Authors: We agree with your consideration. As conservative treatment failure did not correlate with age in the multivariable logistic regression model, in the new version of the manuscript, we abstained from any assumption on the role of age on the risk of recurrence. However, we limited our discussion to what has already been reported in the literature. Thank you.

"The role of age should be investigated. In our study, the mean age difference between patients for whom conservative treatment failed and those in which it was successful was 6.96 years, suggesting the hypothesis that failure correlates with age at the univariable analysis. However, based on the data available and the design of this study it is not possible to establish whether the assumption that failure correlates with age due not to a more aggressive clinical presentation in the acute episode [2] but a higher recurrence rate is valid. In fact, our study is not powered by an adequate statistical analysis for the outcome recurrence."

The manuscript has now been reviewed for minor language errors with the help of a professional editing software (https://app.grammarly.com/ Premium version). Thank you.
